# Exploration of Deformation of F-Actin during Macropinocytosis by Confocal Microscopy and 3D-Structured Illumination Microscopy

**Linyu Xu [1,2], Yanwei Zhang [1,2], Song Lang [1,2] and Yan Gong [1,2,\*]**

[1] Division of Life Sciences and Medicine, School of Biomedical Engineering (Suzhou), University of Science and Technology of China, Suzhou 215163, China; xulinyu0408@outlook.com (L.X.); zhangyw@sibet.ac.cn (Y.Z.); langs@sibet.ac.cn (S.L.)

[2] Suzhou Institute of Biomedical Engineering and Technology, Chinese Academy of Sciences, Suzhou 215163, China

[\*] Correspondence: gongy@sibet.ac.cn

**Abstract:** Since their invention, confocal microscopy and super-resolution microscopy have become important choices in cell biology research. Macropinocytosis is a critical form of endocytosis. Deformation of the cell membrane is thought to be closely related to the movement of F-actin during macropinocytosis. However, it is still unclear how the morphology of F-actin and the membrane change during this process. In this study, confocal microscopy was utilized for macroscopic time-series imaging of the cell membranes and F-actin in cells induced by phorbol 12-myristate 13-acetate (PMA). Super-resolution structured illumination microscopy (SIM), which can overcome the diffraction limit, was used to demonstrate the morphological characteristics of F-actin filaments. Benefiting from the advantages of SIM in terms of resolution and 3D imaging, we speculated on the regular pattern of the deformation of F-actin during macropinocytosis. The detailed visualization of structures also helped to validate the speculation regarding the role of F-actin filaments in macropinocytosis in previous studies. The results obtained in this study will provide a better understanding of the mechanisms underlying macropinocytosis and endocytosis.

**Keywords:** F-actin; macropinocytosis; 3D-SIM; confocal microscopy; morphological study





## 1. Introduction

Endocytosis is a dynamic biological process in which cells internalize various molecules. The membrane invaginates or protrudes out pseudopodia during the endocytosis process, forming an Ω-shaped membrane profile and Ω-profile fission [1]. Endocytosis is key to the normal functioning of cellular activities.

Several different types of endocytosis in eukaryotic cells exist, ranging from receptor-mediated uptake of soluble ligands by clathrin-coated vesicles to ingestion of large particles by phagocytosis [2]. Macropinocytosis is a critical form of endocytosis discovered and named by Warren Lewis in 1931 [3]. It is preceded by vigorous plasma membrane activity, and the cell transports medium into primary endocytic vesicles through the formed ruffles and cups of the plasma membrane [4–7]. Swanson et al. elaborated on the entire process of macropinocytosis, from ruffle formation, ruffle closure and cup closure to the formation of macropinosome vesicles [4]. Macropinosomes are usually 0.2 to 10 μm in diameter, larger than other pinocytic vesicles [8]. Macropinocytosis is a crucial physiological process. Some cells, such as T-cells and macrophages, use macropinocytosis for feeding [9,10]. Moreover, many cancers utilize macropinocytosis as a small-molecule supplement [11,12]. When cancer cells are in the presence of limited nutrients from small molecules, they also use macropinocytosis to consume necrotic cell debris [13]. Increasing our understanding of macropinocytosis may reveal new methods and targets for cancer therapy [14,15]. This

process is also involved in immune responses [16] and is vital for immune surveillance [17]. In addition, macropinocytosis may be a backdoor used by some viruses, including vaccinia and Ebola, to enter cells [18,19]. It is also likely the primary site where vaccine mRNAs are expressed [20].

Actin filaments can act as mechanical elements that drive cell movement or undergo shape changes [21]. It is known that actin provides forces for endocytosis in yeast [22] and is crucial for all kinetically distinguishable forms of endocytosis in mammalian cells [23]. Macropinocytosis relies heavily on the spontaneous activation of the signaling and cytoskeleton networks, which drive cup formation independently of the physical template or surface receptor [24,25]. Among the actin-binding proteins, coronin forms distinctive 'crowns' on the dorsal surface of cells which have now been identified as macropinocytic cups [26,27]. Actin polymerization mediated by the Arp2/3 complex and its activators, SCAR/WAVE and WASP, is essential for the formation of macropinosomes, and formin-directed actin polymerization also contributes to this process [28–30]. Formin G is highly enriched in the cups of macropinosomes and acts as an actin polymerase of Arp2/3-nucleated filaments to allow efficient membrane expansion and engulfment of extracellular material [31]. However, although biochemical methods indicate that F-actin may provide force to bend the membrane, it is still unclear how the morphology of F-actin and the membrane change and interact during macropinocytosis. The elucidation of the macropinocytosis mechanism is mainly based on theoretical models rather than actual images.

The size of microfilaments is in the order of tens of micrometers. Due to the diffraction limit of visible light, traditional light microscopy and confocal microscopy are still ineffective for exploring the cytoskeleton and membrane of cells, with a spatial resolution higher than the subcellular level. In addition, the lack of 3D structural information is an important reason for the inability to confirm the mechanism of macropinocytosis. Super-resolution fluorescence microscopy, which achieves spatial imaging resolution beyond the resolution limit, has been widely applied in life sciences. Structured illumination microscopy (SIM) is a super-resolution microscopy with high spatiotemporal resolution, enhanced 3D imaging capability, and low phototoxicity [32,33]. SIM was used to discover endocytosis and phagocytosis—from uptake at the plasma membrane, endocytic coat formation, and cytoskeletal rearrangements to endosomal maturation [34]. However, little work has been done on super-resolution imaging and 3D imaging of F-actin during the formation of macropinosomes. Detailed visualization will provide a better understanding of how cells complete macropinocytosis and ingest extracellular materials.

We proposed a fast and sensitive three-dimensional SIM based on asymmetric three-beam interference (ATI-SIM) in 2021 [35]. New polarization modulation and rapid acquisition methods were adopted in the ATI-SIM. Compared to traditional SIM, ATI-SIM has a simple structure, high speed, and good stability. Commercial confocal microscopy and ATI-SIM were utilized in this study to perform high-resolution and super-resolution imaging of the cytoskeleton and cell membrane structures during macropinocytosis. Super-resolution images can intuitively and powerfully explain how F-actin filaments undergo morphological changes and form bulges consistent with the structure of macropinosomes. The results of this study provide visual evidence of the complex dynamic structural changes in F-actin during macropinocytosis. This suggests that studies and interventions on cytoskeletal movements may have important implications for research on macropinocytosis and endocytosis. Furthermore, the work presented in this paper can further demonstrate the application of ATI-SIM in related biomedical fields.

## 2. Materials and Methods

### 2.1. Brief General Description

It is well known that the advantage of confocal microscopy is the vast imaging field, whereas SIM has superior high resolution. The strengths of both confocal microscopy and ATI-SIM were utilized in this study. First, commercial confocal microscopy was used to perform time-series high-resolution imaging of pre-treated cell samples over a large field of

view. This confirmed the ability of phorbol ester to induce macropinocytosis in SW480 cells. The observed structural changes in macropinosomes on the cell membrane also provided a reference for subsequent super-resolution experiments. Next, based on ATI-SIM, we performed super-resolution imaging of F-actin in 2D and 3D modes. We explored the structural characteristics of F-actin before and after macropinosome formation.

### 2.2. Apparatus

In this study, self-developed structured illumination microscopy was utilized to observe the fine structure of the cells. In contrast to traditional SIM, it is a fast and sensitive three-dimensional structured illumination microscopy based on asymmetric three-beam interference between $-2$, $+1$, and $+2$ order diffracted light. A segmented achromatic half-wave plate that could modulate the polarization of light without energy loss or time delay was applied in the ATI-SIM. Owing to the asymmetric three-beam interference characteristics, a new acquisition method was adopted in the ATI-SIM which could shorten the acquisition time of each raw image by nearly half. A 60X/1.49NA oil immersion objective was used. The resolution of ATI-SIM can reach up to 115 nm laterally and 309 nm axially, and the speed of three-dimensional imaging by ATI-SIM is 30% faster than that of traditional SIM [35]. Figure 1 shows a simplified diagram and photograph of ATI-SIM.

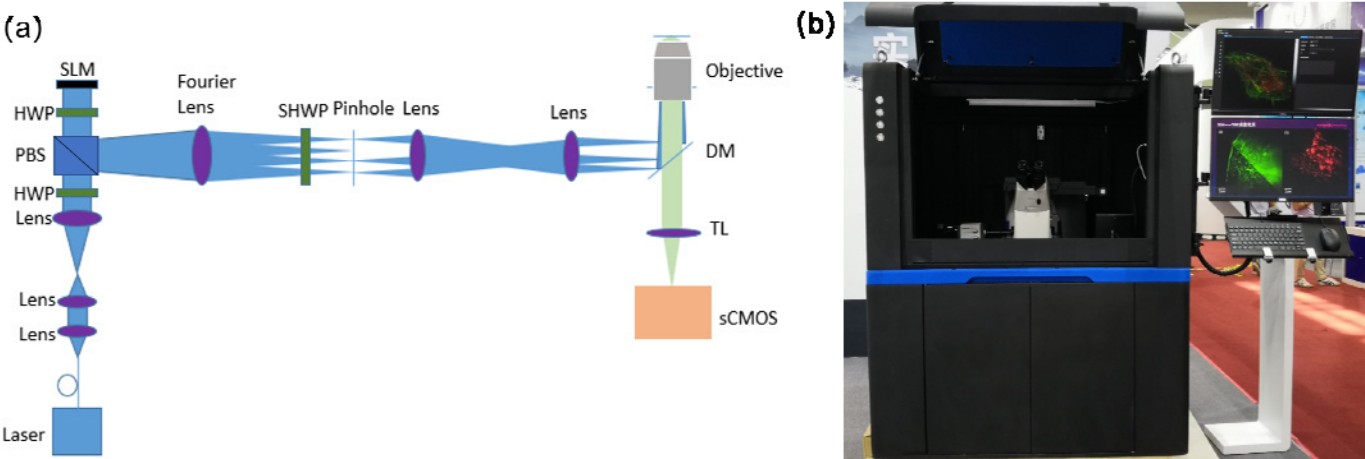

**Figure 1.** (**a**) Simplified diagram of ATI-SIM; (**b**) photo of ATI-SIM.

Cells were also imaged using an A1R HD25 confocal microscope (Nikon, Japan) in the experiments. A 100X/1.45NA oil immersion objective was used.

### 2.3. Cell Culture

SW480 cells were purchased from the cell bank of the type culture collection of the Chinese Academy of Sciences (Shanghai, China) and verified by short tandem repeat assays for their identification. The cells were cultured in DMEM (Thermo Fisher Scientific, Waltham, MA, USA) supplemented with 10% fetal bovine serum (Gibco, Invitrogen, Waltham, MA, USA), 100 µg/mL streptomycin, and 100 U/mL penicillin (Sigma-Aldrich, St Louis, MO, USA) on 10 cm plates at 37 °C in the presence of 5% $CO_2$.

### 2.4. Reagents and Chemicals

Actin-stain™ 488 and 555 phalloidin were purchased from Thermo Fisher Scientific. 1,1′-Dioctadecyl-3,3,3′,3′-tetramethylindocarbocyanine perchlorate (Dil) and phorbol 12-myristate 13-acetate (PMA) were purchased from Beyotime.

*2.5. Cell Treatment and Microscopy Imaging*

2.5.1. Confocal Microscopy Imaging

Diacylglycerol (DAG), phorbol 12-myristate 13-acetate (PMA), and oncogenic transformations are known activators of macropinocytosis [4]. PMA was used to induce macropinocytosis in this study. The cells were cultured in glass dishes for 24 h, then treated with 0.1 µM PMA for 0/5/10/20 min. The cells were fixed with 4% paraformaldehyde in PBS for 20 min and permeabilized with 0.1% Triton for 10 min at room temperature. For F-actin staining, cells were treated with Actin-stain™ 488 phalloidin for 30 min. For cell membrane staining, cells were treated with 1,1′-dioctadecyl-3,3,3′,3′-tetramethylindocarbocyanine perchlorate (DiI) for 30 min at room temperature. The cells were then washed three times with PBS for 5 min.

The A1R HD25 confocal microscope(Nikon, Japan) was used to observe in time series from a macroscopic field of view to demonstrate the induction effect of PMA on the SW480 cells. The channel, mode, exposure, excitation, and luminousness were set for optimum imaging. This provided a basis for subsequent super-resolution microscopy imaging.

2.5.2. ATI-SIM Imaging

The resolution of confocal microscopy is insufficient to observe the details of the F-actin structure. ATI-SIM was used to perform super-resolution imaging of F-actin in normal and pre-induced cells. According to the confocal microscopy imaging results, the completed vesicles formed on the cell membrane after 20 min of PMA treatment. Therefore, we selected cells pre-treated with PMA for 20 min for super-resolution imaging. The cells were cultured in glass dishes for 24 h, then treated with no PMA and 0.1 µM PMA for 20 min respectively. The cells were fixed with 4% paraformaldehyde in PBS for 20 min and permeabilized with 0.1% Triton for 10 min at room temperature. For F-actin staining, cells were treated with Actin-stain™ 555 phalloidin for 30 min. The cells were then washed with PBS for 5 min (three times). Cells were imaged by the 2D-SIM mode and 3D-SIM mode.

**3. Results**

*3.1. Time-Series Imaging by Confocal Microscopy*

To visualize the formation of macropinosomes in the membranes of SW480 cells, the A1R HD25 confocal microscope(Nikon, Japan) was used to observe the biological samples fixed after treatment with the same dose of PMA for 0, 5, 10, and 20 min. In Figure 2, red fluorescence indicates the cell membrane, and green fluorescence indicates F-actin. The structures of interest are inside the white box. In the absence of induction, endocytic vesicles were not present on the cell membrane (Figure 2a,b,d), showing that cup-like structures were formed on the surface of the cell membrane after being pre-treated with PMA. With an increase in action time, the shape of the cup-like protrusions became more apparent. After PMA treatment for 20 min, the bottom of the cup-like protrusions contracted, and relatively complete round vesicles formed on the cell membrane (Figure 2d). Limited by the resolution of confocal microscopy and the intensity of green fluorescence, the location of F-actin can only be roughly determined. As indicated by arrows in Figure 2b–d, it can be observed that green fluorescence representing F-actin appeared inside the vesicles formed by the cell membrane. This means that F-actin also underwent morphological changes and aggregated inside the cup-shaped protrusions of the cell membrane. The results were similar after repeated experiments.

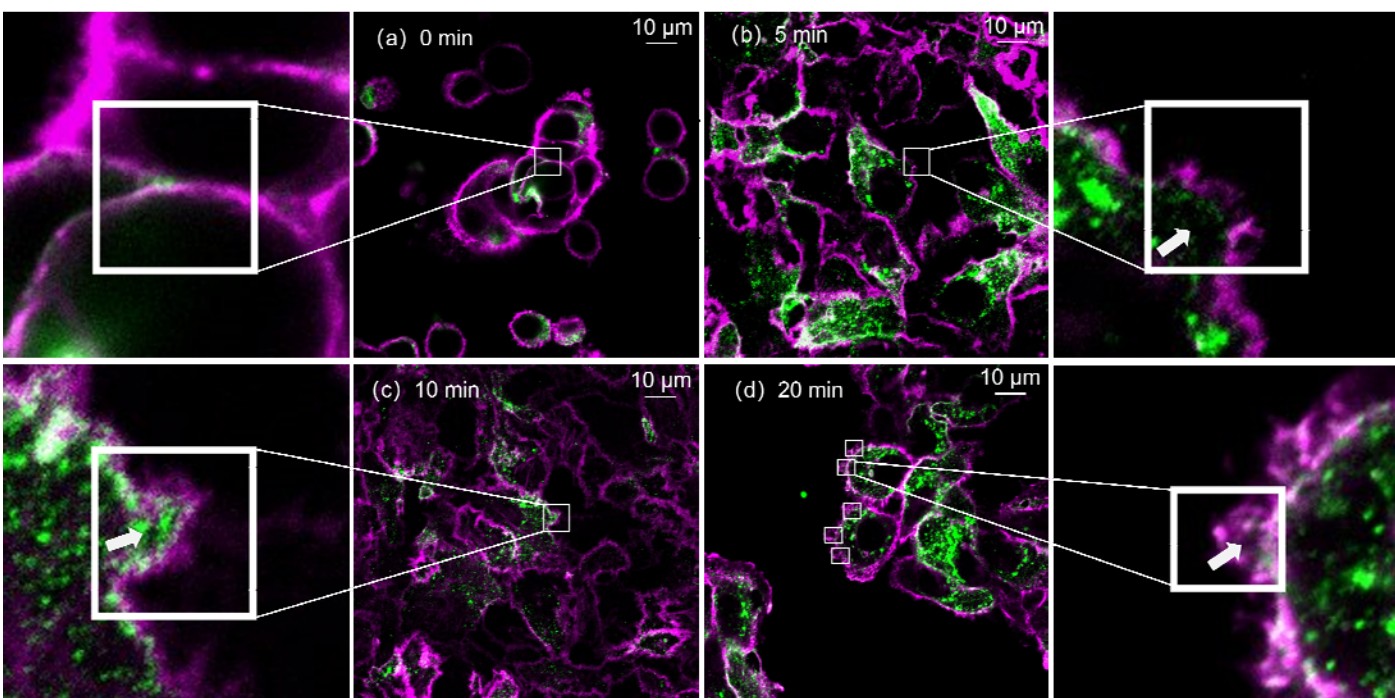

**Figure 2.** Cell membrane (red) and F-actin (green) fluorescence imaging by confocal microscopy. Cell membrane was stained with DiI, and F-actin was stained with phalloidin 488. The treatment times of PMA on cells were 0, 5, 10, and 20 min in (**a**–**d**), respectively.

Time-series imaging by confocal microscopy confirmed that PMA induced deformation of the membrane and F-actin in SW480 cells, and macropinosomes were generated on the cell surface. However, the fine structure of F-actin could not be observed. Morphological changes in F-actin and its role in the formation of macropinosomes could not be visualized. To resolve this, we performed super-resolution imaging of F-actin before and after macropinocytosis using ATI-SIM.

### 3.2. Super-Resolution Imaging by ATI-SIM

ATI-SIM was used to characterize and analyze the fine structures of F-actin in the process of macropinocytosis.

First, super-resolution imaging of F-actin in uninduced cells was performed by 2D-SIM. During the physiological activities of cells, the cytoskeleton is always in the process of dynamic change. Thus, various morphologies of F-actin exist in cells in different states. However, it is impossible to distinguish the different types of structures of F-actin in Figure 2. In this study, super-resolution imaging of F-actin in uninduced SW480 cells was performed by ATI-SIM in 2D mode. After imaging different individual SW480 cells, we summarized the three primary microfilament morphologies in SW480 cells, as shown in Figure 3. F-actin is elongated, running through the entire cell, as shown in Figure 3a. This is the most common morphology observed in other studies. In previous studies, it was found that such microfilaments traversing the ventral side of the cell gradually disappeared during macropinocytosis [36]. As shown in Figure 3b,c, F-actin was relatively concentrated at the cell edge, and these morphologies were rarely observed in macropinocytosis-related studies. In contrast, the structures of F-actin in the middle of the cells were not apparent. In Figure 3c, the microfilaments are in the shape of long strips and are intertwined and overlapped to form a bird's nest-like shape. These imaging results serve as a reference for analyzing F-actin transformations.

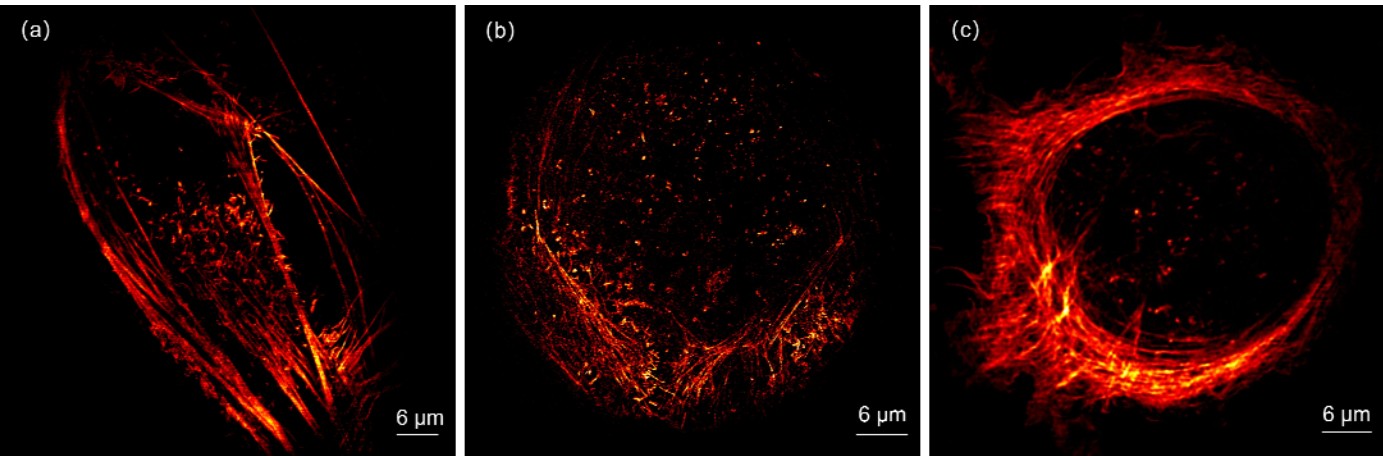

**Figure 3.** Super-resolution imaging of F-actin in uninduced cells by ATI-SIM; (**a–c**) show the main morphologies of microfilaments observed in imaging experiments. F-actin was stained with phalloidin 555.

The above confocal imaging experiments proved that macropinocytosis occurred in SW480 cells after 20 min of PMA treatment. Therefore, SW480 cells were still pre-treated with PMA for 20 min to induce macropinocytosis, and the F-actin of these pre-treated cells was visualized by ATI-SIM. Figure 4 shows multiple sets of 2D and 3D super-resolution images of F-actin morphology during macropinocytosis. It shows different typical F-actin morphologies in the process of macropinocytosis. Figure 4d–f are 3D super-resolution images with depths of 1.8 μm, 1.6 μm, and 0.9 μm, respectively. Figure 4a–c are 2D super-resolution images of the intermediate layers in Figure 4d–f, respectively. Figure 4g shows each layer of the three-dimensional structure in the box shown in Figure 4e. Figure 4h shows each layer of the vesicular three-dimensional structure formed by the microfilaments inside the cell in Figure 4f. Compared with Figure 3, the morphology of F-actin changes in the same trend. (1) As shown in Figure 4a,d, F-actin in the middle part of the cells depolymerized and was no longer filamentous; it became short, messy, and clustered toward the edges of the cells. (2) As shown in white boxes in Figure 4, short and numerous filaments agglomerated together to form multiple protrusions; these protrusions were arranged around a quasi-circular arrangement formed by inner microfilaments. (3) As shown in Figure 5g, the structure of each layer of the bubble-like three-dimensional body formed by the microfilaments is different; from the basal surface to the largest cross-section of the macropinosome, the change trend of cellular filaments is from sparse to dense. This indicates that, inside the vesicle, the microfilaments support the cell membrane in a specific solid three-dimensional structure rather than a sheet-like structure or a ring-like structure. (4) As shown in Figure 4f,h, three-dimensional bubble-like structures appeared inside the cells. They are circular in cross-section in both the xy and xz planes. The hollow spherical structures can be considered as the initial structures of macropinosomes. By comparing with Figure 2, we found that the three-dimensional bulges formed by F-actin were consistent with the macropinosomes formed by the cell membrane.

Five pinocytic vesicles formed by the cell membrane are shown in Figure 5a. The maximum dimensions in the xy plane that could be measured using IMAGEJ software (Version 1.53r, National Institutes of Health, Bethesda, MD, USA) were 2.7 μm, 4.22 μm, 2.22 μm, 2.89 μm, and 2.8 μm. The numerous structures in Figure 5b–d are the bulging structures formed by F-actin toward the outside of the cells. The white dashed part is a curve simulated to represent the outside of F-actin filaments in the regions where macropinosomes were not produced. We measured the maximum dimension in the xy plane from the white dashed line to the outermost end of the raised structure using IMAGEJ software (Version 1.53r, National Institutes of Health, Bethesda, USA). The size of these structures was 2–3.6 μm. The dimensions of the five cell membrane vesicles

in Figure 5a and the 14 microfilament bulging structures in Figure 5b–d are shown in Figure 5e. It was considered that the size of the F-actin structures protruding toward the outside of cells was consistent with the size of macropinosomes formed by the cell membrane.

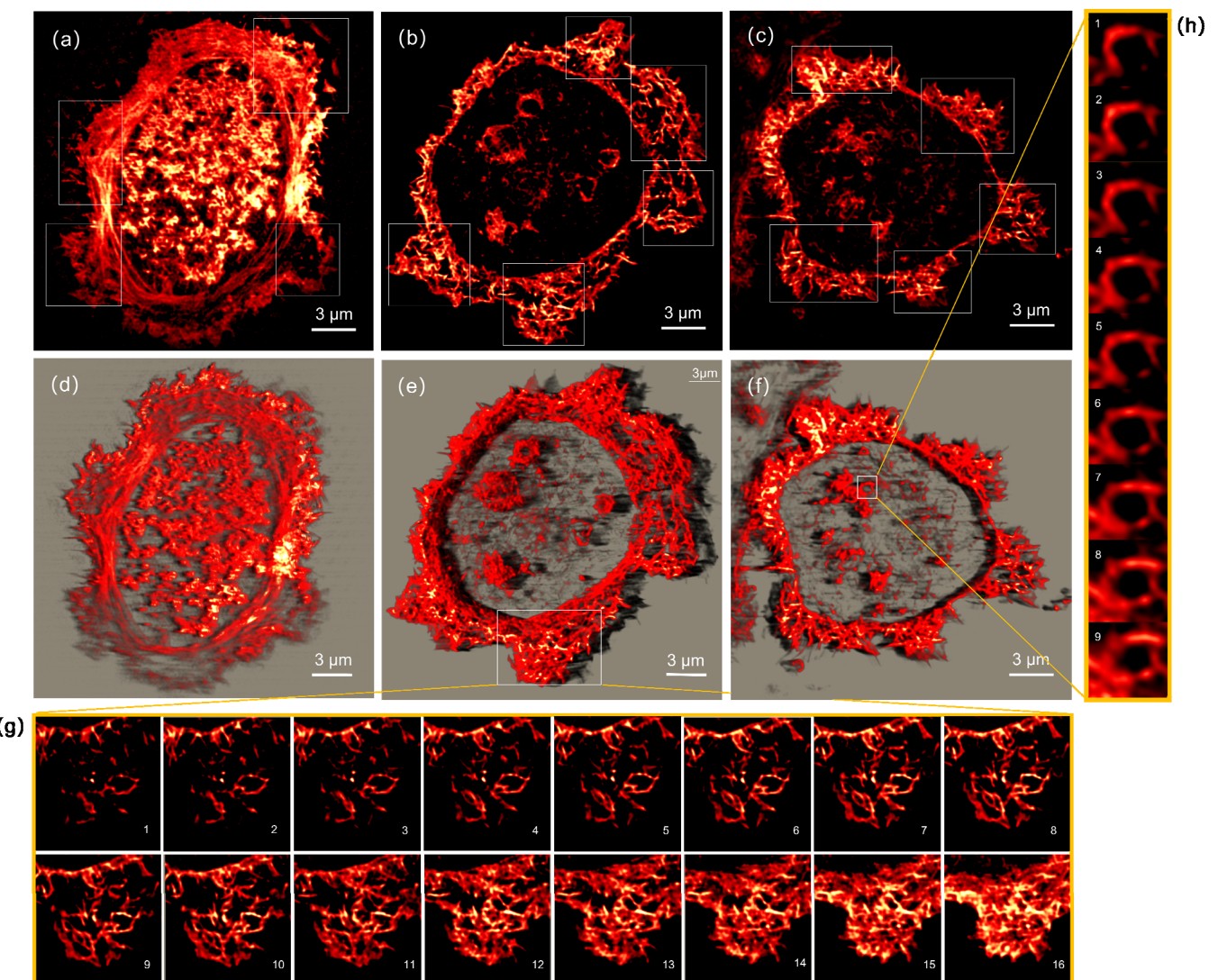

**Figure 4.** Super-resolution imaging of F-actin during macropinocytosis by ATI-SIM. Cells were pretreated with PMA for 20 min; (**a–c**) 2D-SIM reconstructed image of F-actin; (**d**) 3D-SIM reconstructed image of F-actin with 1.8 μm depth corresponding to (**a**); (**e**) 3D-SIM reconstructed image of F-actin with 1.6 μm depth corresponding to (**b**); (**f**) 3D-SIM reconstructed image of F-actin with 0.9 μm depth responding to (**c**); (**g**) each layer of the 3D structure in the box in (**e**); (**h**) each layer of the vesicular 3D structure formed by the microfilaments inside the cell in (**f**). F-actin was stained with phalloidin 555.

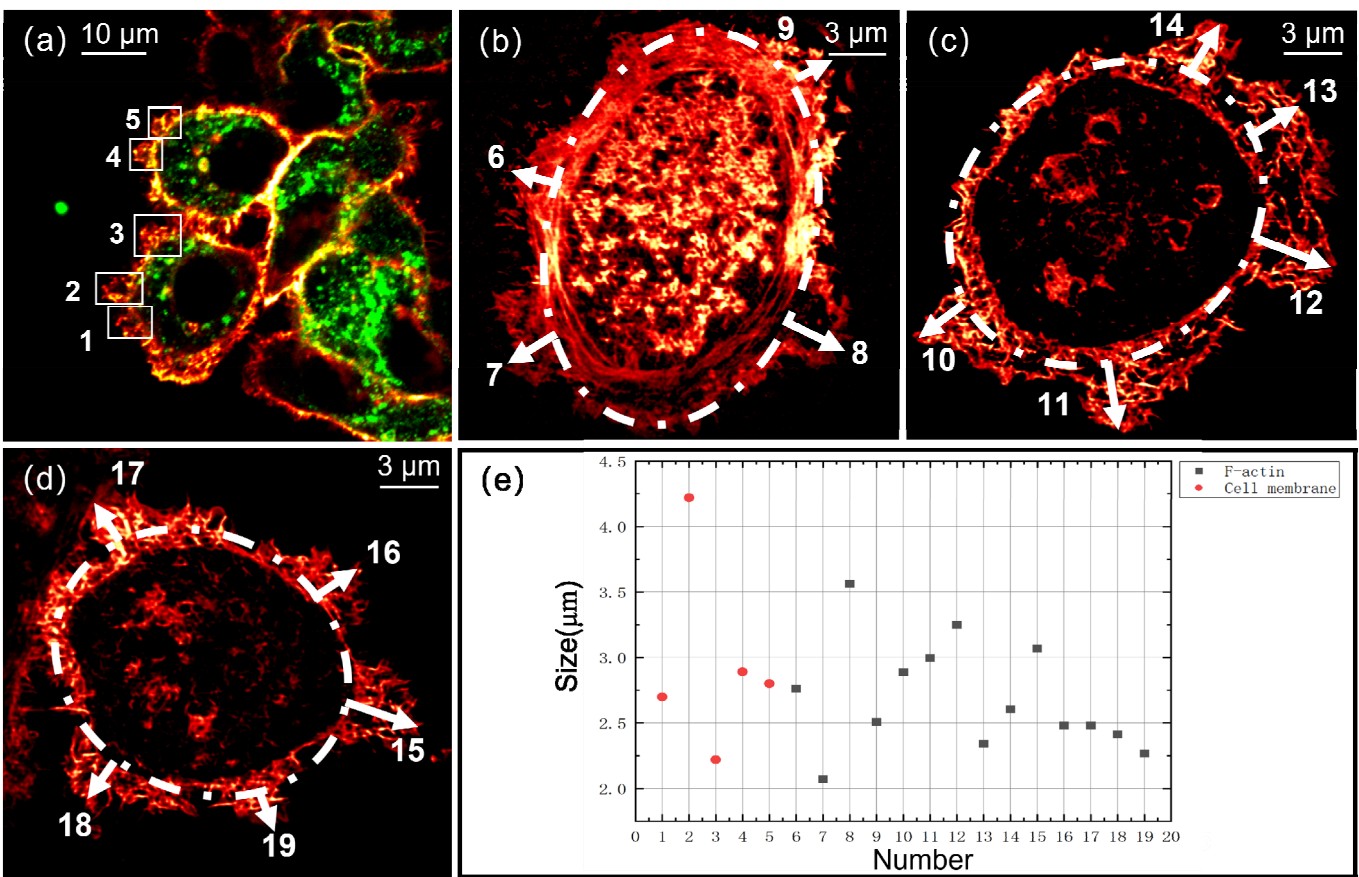

**Figure 5.** Schematic illustration of the size measurements in the xy plane of raised structures formed by the cell membrane and F-actin during macropinocytosis; (**a**) the raised structures formed by the cell membrane; (**b–d**): the raised structures formed by F-actin; (**e**) statistics of the dimensions of the raised structures. The cell membrane was stained with Dil, and F-actin was stained with phalloidin 555.

## 4. Discussion

An in-depth study of macropinocytosis has revealed that this process plays a crucial role in cells. Due to the lack of understanding of the macropinocytosis process and the formation mechanism of macropinosomes, there is no definite classification or differentiation of this mode of endocytosis. The present study showed that dynamic changes in F-actin are closely related to macropinocytosis. However, most studies on the relationship between the cytoskeleton and macropinocytosis require more intuitive and valid evidence to support speculation. Moreover, the inadequacy of imaging methods for the fine structure of F-actin and cell membranes largely limits further study and understanding of the macropinocytosis process.

In this study, we mainly investigated the regular pattern of F-actin deformation in macropinocytosis using confocal microscopy and ATI-SIM. Because of the vastly different modes of macropinocytosis occurring in various cells, we first performed time-series imaging of PMA-treated SW480 cells using confocal microscopy with a large field of view. The process of cell membrane generation in macropinosomes and F-actin undergoing deformation were observed. PMA induced macropinocytosis in SW480 cells. Relatively complete vesicles were formed in cells pre-treated with PMA for 20 min. In the process of macropinocytosis, F-actin tends to aggregate near the cell membrane and can also be observed in macropinosomes.

Based on the above experiments, ATI-SIM was used to perform super-resolution imaging of F-actin in untreated cells and cells pre-treated with PMA for 20 min to observe the deformation of F-actin more clearly. Through comparison, we concluded that, when

macropinocytosis occurs, F-actin in SW480 cells undergoes the following changes: the original long microfilaments are broken or depolymerized; they gradually aggregate in the vicinity of the cell membrane, and the short, messy filaments clump together to form multiple raised globular structures toward the outside of the cell; and some empty stereoscopic vesicles appear inside cells. As these raised globular structures were consistent in shape and size with the macropinosomes formed by the cell membrane, we speculate that macropinosomes form when the cell membrane is pushed by deformed F-actin. In previous visualizations of actin dynamics during macropinocytosis, it was observed that F-actin surrounding the macropinocytic cup increased and that the process occurred by formation of a small actin-rich outward protrusion [37]. It was suggested that macropinocytosis could appear where F-actin was enriched and that F-actin running through the cells would reorganize into fragments during macropinocytosis [36]. It was also observed that an empty void appeared (a presumptive macropinosome) in the subcortical space under GFP-LifeAct-labeled tent pole ruffles at the point of tent pole crossover [38]. The results in this paper are consistent with those in the references and provide more spatial and accurate evidence for the related hypothesis.

In this study, the morphological changes in F-actin and the cell membrane during macropinocytosis were explored using advanced visualization methods. The study combined the advantages of confocal and super-resolution microscopies. This again demonstrates that high-/super-resolution microscopy can be a powerful tool for endocytosis-related research and has important application value in life science.

**Author Contributions:** Conceptualization, L.X.; methodology, L.X.; software, Y.Z.; validation, L.X., Y.Z. and S.L.; formal analysis, L.X.; investigation, L.X.; resources, Y.G.; data curation, L.X.; writing—original draft preparation, L.X.; writing—review and editing, Y.G.; project administration, Y.G. All authors have read and agreed to the published version of the manuscript.

**Funding:** This research was funded by the National Natural Science Foundation of China (62005307 and 61975228), "20 articles" of colleges and universities in Jinan (2019GXRC0420), the Scientific Research and Equipment Development Project of the Chinese Academy of Sciences (YJKYYQ20210031 and YJKYYQ20180032), and the Jiangsu Province Innovation and Entrepreneurship Talent Project.

**Institutional Review Board Statement:** Not applicable.

**Informed Consent Statement:** Not applicable.

**Data Availability Statement:** Not applicable.

**Conflicts of Interest:** The authors declare no conflict of interest.

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
