# Peer review of "Exploration of Deformation of F-Actin during Macropinocytosis by Confocal Microscopy and 3D-Structured Illumination Microscopy"

_photonics, doi:10.3390/photonics9070461_

Round 1

Reviewer 1 Report

The manuscript by Linyu Xu, Yanwei Zhang, Song Lang, and Yan Gong, entitled "Exploration of deformation of F-actin during macropinocytosis by confocal microscopy and 3D-structured illumination microscopy ", presents a comprehensive work of optical microscopy to investigate further the processes and the formation mechanisms leading to macropinocytosis.

Macropinocytosis is a crucial endocytosis mechanism exploited by the cells for many purposes, and it has essential physiological roles. Namely, it can be used as for feeding the cell itself, or to consume necrotic cell detritus, or to uptake soluble proteins, and it seems to be a primary site where mRNAs vaccines are expressed.

Since it is still unclear how cytoskeletal proteins change during macropinocytosis vesicle formation, the authors addressed this question by using confocal microscopy and with 3D-Structured Illumination Microscopy for its capability to overcome the diffraction limit.

The experiments consisted in preparing the SW480 cell line and treating them with phorbol 12-myristate 13-acetate (PMA) for 20 minutes to induce macropinocytosis while performing time-series imaging at 0, 5, 10, and 20 minutes. To do so they stained F-actin and the cell membrane.

Thanks to these two microscopy techniques, the authors observed that, upon PMA treatment, the F-actin filament underwent some morphological modifications. In particular, from the long micro filamentous shape, they get broken or polymerized and aggregated close to the cell membrane. After analyzing shape and size, the authors speculate that the macropinosome could be formed when the cell membrane gets pushed by the deformed F-actin.

The document is clearly written and easy to read. The method description and the results' discussion are easy to follow also for a person not an expert in the field. 

The use of confocal and 3D SIM to evaluate the modification of F-actin is interesting, and it could be helpful in the field related to macropinocytosis studies. I believe that the manuscript fits into the MDPI Photonics journal topics, and I am positive that the results and methods presented will be appreciated by the journal readers.

I believe that the manuscript by Xu L. And collaborators deserves to be published in the MDPI Photonics journal after few revisions:

  • I would correct some grammar for example changing row 104 not to use the word “and” at the beginning of the phrase.Then in line 368 there is a typos in the title of the cited article "on asymmetric…”. I suggest the authors to carefully review the manuscript and to correct eventual typos. 
  • References 2, 5 has a typos in the title “micropinocytosis”. The same word is found in the text in line 136 and 199, but as far as I know Macropinocytosis is actin-dependent and Micropinocytosis is actin-independent therefore maybe is a typos? If these are not typos I would suggest to explain better the concepts
  • As a general comment and just a suggestion that the authors can adopt or discard, in my opinion, I would avoid redundancy in the text. For example, lines 83-84 say the same as lines 98-99, of lines 186-187, 267-268, and 287-288. I understand that the authors want to stress the goal of the work, but I feel it should not be repeated so much or at least written differently.
  • Redundancy can also be found in lines 136-139 and in lines 153-156.
  • Fig.2a, for clarity, I would add a zoomed white box showing the reader the absence of the structure of interest.
  • In Fig 2d, I would indicate the microfilaments aggregates with arrows; then, I would write it in lines 173-174 to make it more accessible for persons unfamiliar with these structures. 
  • In lines 220-221, I would re-write the sentence to clarify that the F-actin distribution pattern radically changes from 4a to 4c and 4e, and I would better explain why it is so different (is it for a different PMA treatment timing?). I would also write more in detail the caption of figure 4 because it is not clear if, in 4a-4c-4e, the PMA treatment was the same. It is written it is the same but the cell in 4a and 4b present a completely different distribution pattern compared to the 4c-d-e-f. I would like the authors to better explain these results.
  • It could be even more interesting if, in the discussion part, lines 278-286, there were some references in agreement with the results obtained by the authors.

Author Response

Dear reviewer:

        Thank you for your comments concerning our manuscript “Exploration of deformation of F-actin during macropinocytosis by confocal microscopy and 3D-structured illumination microscopy” (photonics-1749910). These comments are very valuable and we have studied carefully. Based on these advice and suggestions, we have revised the manuscript. The main revised portion are marked in yellow in the manuscript. The respond and manuscript are in the attachment. Please see the attachment.

Reviewer 2 Report

The manuscript by Xu et al reports about a morphological study of deformation of F-actin in SW480 cells during macropinocytosis induced by PMA. The authors used a commercial confocal microscope from Nikon and a home-built three-dimensional SIM setup based on asymmetric three-beam interference (ATI-SIM). The manuscript sets an application example of the ATI-SIM technology developed by the authors earlier.

The manuscript is well written in general, however it lacks some technical details which could be expanded. For example, very little details are provided about the confocal microscope. I suggest to add the details about the objective used for imaging, as it defines the achievable resolution.

Regarding the resolution – the authors state that an ordinary confocal fluorescence imaging is not sufficient for resolving the structure and size of the F-actin protrusions during micropinocytosis. At the same time, the graph 5E states that the average sizes of such protrusions are on the order of a couple of microns. This is definitely above the diffraction limit of confocal microscopy, unless a very poor objective with an extremely low NA was used.

Therefore, the motivation of involvement of the 3D-SIM imaging is not clear to me. The authors rely exclusively on 2D projections for their analysis. It is nice to see that the cell structures are different at the different planes from the coverslip, but is that important for the study?

The language style of the manuscript is good, just few sentences need to be corrected – line 66 (“The size of the cytoskeleton is on the order of tens of micrometers” – cytoskeleton is a general name for a complex, dynamic network of interlinking protein filaments, each of which has a different size) and two first sentences of the section 2.1.

The images are lacking colormaps. Red/green combination of colors for two-color images is not CVD-friendly (color vision deficiency). Please consider substitute colors based on https://doi.org/10.1364/OE.21.009862 and https://doi.org/10.1038/d41586-021-02696-z . 

Author Response

(The authors gave the same response as above.)

Round 2

Reviewer 2 Report

While the authors answered some of the secondary criticism points, they decided to totally ignore the main question about the method selection - why 3D SIM is needed for answering the application question.

Author Response

Dear reviewer:

       Thank you for your comment concerning our manuscript  “Exploration of deformation of F-actin during macropinocytosis by confocal microscopy and 3D-structured illumination microscopy” (photonics-1749910). To answer the question why 3D SIM is needed, we give the following answers and modified parts of the article are marked in green.

  1. Currently, there are different conjectures and inferences on the mechanism of micropinocytosis [1][2]. Most of these models are summarized from two-dimensional confocal imaging images. The lack of 3D structural information is an important reason for the inability to judge the accuracy of these inferences. In recent years, progress in theoretical and computational approaches has allowed the dynamical properties of cellular and sub-cellular scale membrane deformation, such as amoeboid motion and filopodia formation [3]. Many studies have focused on cases that can be approximated in one- and two-dimensional space. Understanding the full nature of membrane/F-actin deformation in macropinocytosis requires full 3D modeling. Super-resolution 3D imaging can provide more spatial information for studying the mechanism of pinocytosis.
  2. As shown in Figure 4(g), the structure of each layer of the bubble-like three-dimensional body formed by the microfilaments is different. From the basal surface to the largest cross-section of the macropinosome, the change trend of cellular filaments is from sparse to dense. Because these microfilament structures are continuous in space, 3D tomography imaging can show the spatial position and shape change trend of microfilaments. It also indicates that inside the vesicle, the microfilaments support the cell membrane in a specific solid three-dimensional structure, rather than a sheet-like structure or a ring-like structure.

         As shown in Figure 4(h), the bubble-like structure inside the cell is circular           in cross-sections in both the xy and xz planes. Super-resolution 3D                       imaging proved it to be a hollow spherical structure. It is consistent with               the empty void appears (a presumptive macropinosome) in the subcortical           space under GFP-LifeAct–labeled tent pole ruffles at the point of tent pole           crossover in the other related study [4]. It can be considered as the initial             structures of macropinosomes.

         Therefore, 3D super-resolution imaging can provide more accurate evidence support for studying the mechanism of macropinocytosis.

[1]Cosimo Commisso; Jayanta Debnath. Macropinocytosis fuels prostate cancer. CANCER DISCOVERY 2018, 8, 800-802.

[2] Szu-Wei Lee; Yijuan Zhang; Michael Jung; Nathalia Cruz; Basheer Alas; Cosimo Commisso. EGFR-Pak signaling selectively regulates glutamine deprivation-induced macropinocytosis. DEVELOPMENTAL CELL 2019, 50, 381-392.

[3]Nen Saito; Satoshi Sawai. Three-dimensional morphodynamic simulations of macropinocytic cups. ISCIENCE 2021, 24, 103087.

[4] Nicholas D. Condon; John M. Heddleston; Teng‑Leong Chew; Lin Luo; Peter S. McPherson; Maria S. Ioannou; Louis Hodgson; Jennifer L. Stow; Adam A. Wall. Macropinosome formation by tent pole ruffling in macrophages. JCB 2018, 217, 3873-3885.
